# ZTRS: Zero-Imitation End-to-end Autonomous Driving with Trajectory Scoring

## Abstract

End-to-end autonomous driving maps raw sensor inputs directly into ego-vehicle trajectories to avoid cascading errors from perception modules and to leverage rich semantic cues. Existing frameworks largely rely on Imitation Learning (IL), which can be limited by sub-optimal expert demonstrations and covariate shift during deployment. On the other hand, Reinforcement Learning (RL) has recently shown potential in scaling up with simulations, but is typically confined to low-dimensional symbolic inputs (e.g. 3D objects and maps), falling short of full end-to-end learning from raw sensor data. We introduce ZTRS (Zero-Imitation End-to-End Autonomous Driving with Trajectory Scoring), a framework that combines the strengths of both worlds: sensor inputs without losing information and RL training for robust planning. To the best of our knowledge, ZTRS is the first framework that eliminates IL entirely by only learning from rewards while operating directly on high-dimensional sensor data. ZTRS utilizes offline reinforcement learning with our proposed Exhaustive Policy Optimization (EPO), a variant of policy gradient tailored for enumerable actions and rewards. ZTRS demonstrates strong performance across three benchmarks: Navtest (generic real-world open-loop planning), Navhard (open-loop planning in challenging real-world and synthetic scenarios), and HUGSIM (simulated closed-loop driving). Specifically, ZTRS achieves the state-of-the-art result on Navhard and outperforms IL-based baselines on HUGSIM.

## 1 Introduction

End-to-end autonomous driving, which aims to map high-dimensional sensor data into an ego-vehicle trajectory with a neural planner, has emerged as a critical research direction. Unlike modularized approaches, where a privileged planner relies on low-dimensional symbolic inputs (e.g. 3D objects and map information), end-to-end methods avoid cascading errors from perception modules (Hu et al., 2023) and leverage rich semantic cues that privileged planners cannot access (Mu et al., 2024).

Two main paradigms have emerged for training planners: Imitation Learning (IL) and Reinforcement Learning (RL). IL requires human demonstrations for training, while RL requires reliable simulation environments. In current research, RL-based methods can scale with massive simulation data (Cusumano-Towner et al., 2025; Jaeger et al., 2025) and simple rewards (Jaeger et al., 2025). They demonstrate more robustness compared with IL-based counterparts, which often face covariate shift during deployment and rely on human demonstrations that can be either noisy or sub-optimal.

However, RL-based methods are still restricted to symbolic inputs. Scaling RL online with sensor data remains impractical: real-world exploration is unsafe, and large-scale sensor simulation is both costly and difficult to make realistic. For instance, diffusion-based world models (Guo et al., 2025; Agarwal et al., 2025) demand thousands of GPU hours just to approximate the scale of public datasets (Dauner et al., 2024).

To achieve the best of both worlds: preserving rich sensor inputs and leveraging reinforcement learning for robust planning, we propose ZTRS: **Z**ero-Imitation End-to-end Autonomous Driving with **TR**ajectory **S**coring, the first framework that eliminates imitation learning entirely from the end-to-end training pipeline.

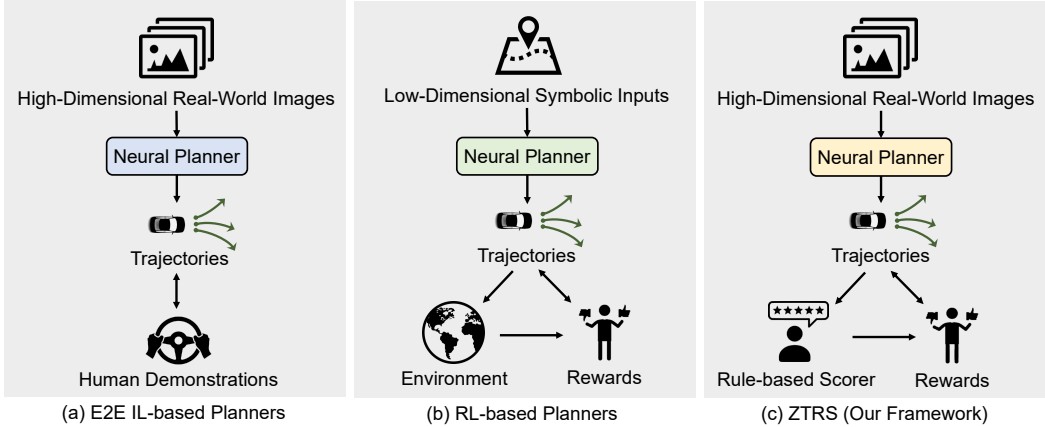

Figure 1: **Comparisons between three paradigms for end-to-end autonomous driving.**

ZTRS builds on three pillars: data, rewards, and policy optimization. For the data pillar, since large-scale sensor data collection is difficult, we rely on offline driving datasets. This naturally transforms our problem into offline reinforcement learning (Levine et al., 2020), where the planner learns to maximize the reward at each data point. For the reward pillar, this offline setting aligns with the open-loop trajectory planning problem (Dauner et al., 2023; Li et al., 2024c; Dauner et al., 2024) in the end-to-end autonomous driving literature, so we adopt the widely-used open-loop planning metrics (Dauner et al., 2024; Li et al., 2025b; Cao et al., 2025) as rewards, enabling efficient evaluation of safety, rule-compliance, and comfort.

The final pillar is policy optimization. Without human demonstrations, policy optimization suffers from the cold-start problem, as random exploration in the continuous trajectory space is highly inefficient (Lillicrap et al., 2015). To address this, we propose Exhaustive Policy Optimization (EPO), a variant of policy gradient designed for offline data and enumerable action spaces. By enumerating a rich trajectory set as the action space, EPO provides dense supervision by optimizing each action in the trajectory set rather than randomly sampled actions. This formulation allows us to efficiently train a high-capacity planner entirely in the offline setting, without imitation learning or additional environment interaction.

We evaluate ZTRS on three autonomous driving benchmarks: Navtest (Dauner et al., 2024) for generic real-world open-loop planning, Navhard (Cao et al., 2025) for planning in challenging real-world and synthetic scenarios, and HUGSIM (Zhou et al., 2024) for simulated closed-loop driving. Our experiments demonstrate that ZTRS exhibits general planning abilities comparable to IL-based planners while maintaining robustness in safety-critical driving situations. Notably, ZTRS establishes a new state-of-the-art on the open-loop planning benchmark Navhard and outperforms IL-based baselines on the closed-loop driving benchmark HUGSIM.

Our contributions are as follows:

1. We introduce ZTRS, a zero-imitation end-to-end autonomous driving framework with trajectory scoring. This framework is the first to solely learn from rewards rather than human demonstrations, while operating fully on high-dimensional real-world images.

2. We propose offline reinforcement learning with Exhaustive Policy Optimization, a variant of policy gradient tailored for enumerable actions and rewards. This optimization process allows us to efficiently train an end-to-end policy from scratch.

3. We evaluate ZTRS on three benchmarks: Navtest, Navhard, and HUGSIM. ZTRS demonstrates strong planning performance compared with other IL-based trajectory scorers under different evaluation protocols (i.e. open-loop planning and closed-loop driving) and diverse sensor data (i.e. real-world images and 3DGS-rendered images). It also achieves the state-of-the-art result on the challenging Navhard benchmark and outperforms IL-based baselines on HUGSIM.

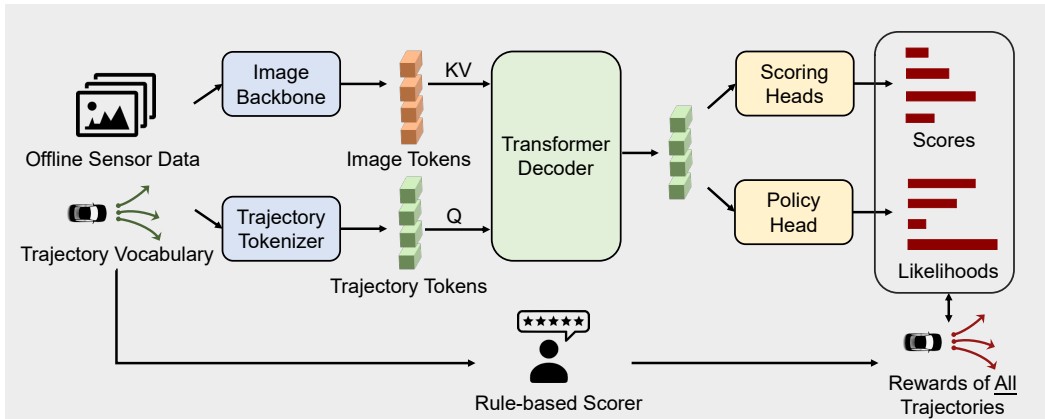

Figure 2: **The Overall Framework of ZTRS.** Given offline sensor data and a fixed set of trajectories, ZTRS first tokenizes these two modalities. In a Transformer Decoder, the trajectory tokens attend to image tokens to acquire the context. Finally, scoring heads and a policy head map the trajectory tokens to rule-based scores and action likelihoods.

## 2 METHODOLOGY

In this section, we elaborate on the framework and the policy optimization technique used in ZTRS.

### 2.1 OVERALL FRAMEWORK

As shown in Fig. 2, ZTRS is a trajectory scorer (Li et al., 2024b; 2025b;e; Wang et al., 2025; Sima et al., 2025; Yao et al., 2025; Li et al., 2025f), whose functionality is to score a discrete set of trajectories $\mathcal{A} = \{a_i\}_{i=1}^n$ instead of regressing to a continuous trajectory. The discrete formulation sidesteps the need for exploring in a large continuous space, also facilitating efficient reward computation in an offline manner.

ZTRS consists of five modules: an image backbone, a trajectory tokenizer, a Transformer Decoder, a policy head, and several scoring heads. The policy head produces probabilities for taking each action in $\mathcal{A}$, while the scoring heads predict the scores for each open-loop metric in the Extended Predictive Driver Model Score (EPDMS, $\mathcal{E}$) (Dauner et al., 2024; Li et al., 2025b; Cao et al., 2025). EPDMS evaluates multiple aspects of driving behavior (e.g., safety, progress, and rule compliance) and can be efficiently computed in an offline manner. Given a state $s$ sampled from an offline dataset $\mathcal{D}$, where $s$ contains sensor data and ego-vehicle status, the forward process involves three steps:

- The image backbone extracts $L$ image tokens $\{x_{img}^i\}_{i=1}^L$ from a frontal-view image, while the trajectory tokenizer encodes trajectory candidates into queries $\{x_{traj}^i\}_{i=1}^n$.

- In the Transformer Decoder, the trajectory queries attend to image tokens.

- The policy head maps the attended trajectory queries $\{x_{traj}^i\}_{i=1}^n$ to probabilities $\pi(\cdot|s)$, and $m$ scoring heads map them to $m$ rule-based scores $\{\mathcal{S}_i(\cdot|s)\}_{i=1}^m$.

During training, the scoring heads are trained with binary classification losses against $\mathcal{E}(s, \cdot)$, while the policy head is trained with Exhaustive Policy Optimization (See Sec. 2.3). At inference, the final trajectory $a \in \mathcal{A}$ is chosen using a weighted average of $m + 1$ scores: $\pi(\cdot|s)$ and $\{\mathcal{S}_i(\cdot|s)\}_{i=1}^m$.

### 2.2 PRELIMINARY: THE POLICY GRADIENT THEOREM

To optimize a trajectory scorer with only rewards, we start with a simplified one-step policy optimization problem where the action space is a finite discrete set $\mathcal{A}$, and $\pi$ is a policy parameterized by $\theta$. In the online RL setting (Sutton et al., 1998), the action $a$ is sampled with $\pi(a|s)$, where $s$ is the current state. In our offline RL setting (Levine et al., 2020), the state $s$ is sampled from an offline

dataset $\mathcal{D}$. Following the notation of Schulman et al. (2015), the policy gradient $g$ is defined as

$$g := \mathbb{E}\left[\Psi(s, a)\nabla_\theta \log \pi_\theta(a \mid s)\right], \tag{1}$$

where the advantage function $\Psi(s, a)$ can represent many quantities, such as the cumulative return or the state-action value function. The gradient can be equivalently written as

$$g = \mathbb{E}\left[\Psi(s, a)\nabla_\theta \log \pi_\theta(a \mid s)\right] \tag{2}$$

$$= \sum_{a' \in \mathcal{A}} \Psi(s, a')\pi_\theta(a' \mid s)\nabla_\theta \log \pi_\theta(a' \mid s) \tag{3}$$

$$= \sum_{a' \in \mathcal{A}} \Psi(s, a')\pi_\theta(a' \mid s)\frac{\nabla_\theta \pi_\theta(a' \mid s)}{\pi_\theta(a' \mid s)} \tag{4}$$

$$= \sum_{a' \in \mathcal{A}} \Psi(s, a')\nabla_\theta \pi_\theta(a' \mid s). \tag{5}$$

This formulation matches the classical Policy Gradient Theorem (Sutton et al., 1999). Notably, the summation over all actions in $\mathcal{A}$ suggests that if the advantage function $\Psi$ can be computed for each action at state $s$, policy optimization can be carried out directly on action likelihoods rather than log-likelihoods, as shown in Eq. 5.

## 2.3 OFFLINE REINFORCEMENT LEARNING WITH EXHAUSTIVE POLICY OPTIMIZATION

When the action space $\mathcal{A}$ is a set of trajectories covering almost all driving possibilities, policy optimization can be formulated as maximizing the objective for an offline dataset $\mathcal{D}$

$$\mathbb{E}_{s \sim \mathcal{D},\, a \sim \mathcal{A}}\left[\Psi(s, a)\right]. \tag{6}$$

This is consistent with the one-step policy optimization problem in Sec. 2.2 if $\Psi$ is derived from open-loop reward signals, which do not require additional environment interaction. This objective can be optimized either in the log-likelihood form on a sampled action (Eq. 1) or in the likelihood form on the entire action space (Eq. 5). The latter formulation,

$$g := \sum_{\substack{a' \in \mathcal{A} \\ s \sim \mathcal{D}}} \Psi(s, a')\nabla_\theta \pi_\theta(a' \mid s) \tag{7}$$

provides much denser supervision for the policy. Eq. 7 also defines our proposed *Exhaustive Policy Optimization (EPO)*, a variant of policy gradient tailored for offline data and enumerable actions. Specifically, EPO optimizes the policy by exhaustively considering every possible action $a \in \mathcal{A}$ and its respective advantage $\Psi(s, a)$. An overview of the pipeline is shown in Fig. 2.

To compute the advantage function $\Psi$, we adopt the EPDMS metric score $\mathcal{E}$ for its efficient and comprehensive evaluation of trajectories. Specifically, $\mathcal{E}$ can cover safety, rule-compliance, and progress for each state-action pair $(s, a)$. The scores $\mathcal{E}(s, \cdot)$ can also be reused for each state $s \in \mathcal{D}$ since the action space is fixed throughout the training process. To further enforce temporal consistency, we subtract a correction term $b(s_t, a_t, a_{t-1}) = \lambda\mathbb{1}\left[\text{EC}(a_{t-1}, a_t)\right]$, where $\lambda$ is a constant, $a_{t-1} = \operatorname*{argmax}_a \pi(a|s_{t-1})$, and EC indicates the violation of Extended Comfort (EC) thresholds (Li et al., 2025b):

$$\Psi(s_t, a_t) = \mathcal{E}(s_t, a_t) - b(s_t, a_t, a_{t-1}). \tag{8}$$

Note that this formulation penalizes inconsistent predictions more strongly than the one used in Li et al. (2025b); Cao et al. (2025). Finally, $\Psi$ is normalized to zero mean and unit variance following Huang et al. (2022); Shao et al. (2024).

## 3 EXPERIMENTS

### 3.1 DATASET AND METRICS

**Dataset.** We conduct experiments on three benchmarks, including Navtest (Dauner et al., 2024), Navhard (Cao et al., 2025), and HUGSIM (Zhou et al., 2024).

Navtest (Dauner et al., 2024) is the evaluation dataset used in NAVSIM. NAVSIM contains 103k and 12k diverse and challenging driving scenarios for model training (Navtrain) and evaluation (Navtest), and introduces simulation-based metrics to better review closed-loop planning capability through open-loop evaluation. During evaluation, the output trajectory is evaluated by a simulator to get rule-based simulation metric scores related to multiple driving aspects, such as traffic rule compliance, comfort, and progress.

Navhard (Cao et al., 2025) further proposes pseudo-simulation on the challenging scenarios in NAVSIM, extending the evaluation to a two-stage paradigm. The first stage follows the original NAVSIM evaluation, while the second stage adopts 3D Gaussian Splatting (3DGS) (Kerbl et al., 2023; Li et al., 2025c) to synthesize subsequent driving scenarios, resulting in 244 initial scenarios and 4164 synthetic scenarios.

HUGSIM (Zhou et al., 2024) is a closed-loop driving benchmark featuring 3DGS-synthesized images. It integrates multiple driving datasets, including KITTI-360 (Liao et al., 2022), Waymo (Sun et al., 2020), nuScenes (Caesar et al., 2020), and Pandaset (Xiao et al., 2021), into a collection of 345 driving scenarios. These scenarios are categorized by difficulty into four levels: easy, medium, hard, and extreme. Specifically, HUGSIM released 49 easy scenarios for regular driving, 126 medium scenarios with inserted vehicles, and 86 hard scenarios as well as 84 extreme scenarios with aggressive vehicles.

**Metrics.** Navtest and Navhard evaluate open-loop planning with EPDMS $\mathcal{E}(s,a)$:

$$\mathcal{E}(s,a) = \left( \prod_{m \in S_{\text{pen}}} m(s,a) \right) \cdot \left( \frac{\sum_{m \in S_{\text{avg}}} w_m \, m(s,a)}{\sum_{m \in S_{\text{avg}}} w_m} \right), \qquad (9)$$

where $s$ is the current state and $a$ is a 4-second trajectory. The penalty metric set $S_{\text{pen}}$ is applied multiplicatively and includes No-at-fault Collisions (NC), Drivable Area Compliance (DAC), Driving Direction Compliance (DDC), and Traffic Light Compliance (TLC), while the weighted metric set $S_{\text{avg}}$ contains Time-to-Collision (TTC), Ego Progress (EP), Lane Keeping (LK), and History Comfort (HC). The extended comfort (EC) from Li et al. (2025b) is also used as a weighted metric to promote temporally consistent driving. $w_m$ is the aggregation weight for metric $m$. Note that human filtering is used in Cao et al. (2025) for calculating $\mathcal{E}$ but not in our training. For HUGSIM, HD-Score is used for closed-loop evaluation. It aggregates Route Completion (RC) with NC, DAC, TTC, HC across an episode of length $T$:

$$\text{HD-Score} = RC \cdot \sum_{t=1}^{T} \left( \prod_{m \in \{NC, DAC\}} m(s_t, \tilde{a}_t) \right) \cdot \left( \frac{\sum_{m \in \{TTC, HC\}} w_m \, m(s_t, \tilde{a}_t)}{\sum_{m \in \{TTC, HC\}} w_m} \right), \quad (10)$$

where $\tilde{a}$ is the ego-vehicle acceleration and steering angle transformed from a trajectory.

## 3.2 IMPLEMENTATION DETAILS

All our models are trained on the Navtrain split with 24 NVIDIA A100 GPUs, while the synthetic data from Navhard and HUGSIM are not used for training. Models are trained for 15 epochs with a total batch size of 528, using a learning rate and weight decay of $2 \times 10^{-4}$ and 0.0. The frontal view with center-cropped front-left and front-right views are concatenated as the input image, which is then resized to $512 \times 2048$. The hyperparameter $\lambda$ in the correction term $b$ is set to 0.2. By default, the action space used in our method has 16384 trajectories, each spanning 4 seconds at 10Hz. These trajectories are obtained through K-means clustering on the nuPlan dataset (H. Caesar, 2021). Following Li et al. (2024b; 2025b); Yao et al. (2025), we default to use the DD3D-pretrained (Park et al., 2021) V2-99 (Lee et al., 2019) as the image backbone in our experiments. The ViT-L Dosovitskiy et al. (2020) backbone is pretrained from Depth-Anything (Yang et al., 2024).

## 3.3 QUANTITATIVE RESULTS

Table 1 reports performance on the challenging Navhard benchmark. Compared with IL-based methods, ZTRS achieves superior scores across most safety and comfort metrics, attaining the highest overall EPDMS (45.5%) with the V2-99 backbone. Further, Tab. 2 shows performance on

Table 1: **Performance on the Navhard Benchmark.** PDM-Closed uses ground-truth symbolic inputs for planning, while other methods rely on sensor data.

| Method | IL | Backbone | Stage | NC | DAC | DDC | TLC | EP | TTC | LK | HC | EC | EPDMS |
|---|---|---|---|---|---|---|---|---|---|---|---|---|---|
| PDM-Closed (Dauner et al., 2023) | ✗ | - | Stage 1 | 94.4 | 98.8 | 100 | 99.5 | 100 | 93.5 | 99.3 | 87.7 | 36.0 | 51.3 |
| | | | Stage 2 | 88.1 | 90.6 | 96.3 | 98.5 | 100 | 83.1 | 73.7 | 91.5 | 25.4 | |
| LTF (Chitta et al., 2022) | ✓ | ResNet34 | Stage 1 | 96.2 | 79.5 | 99.1 | 99.5 | 84.1 | 95.1 | 94.2 | 97.5 | 79.1 | 23.1 |
| | | | Stage 2 | 77.7 | 70.2 | 84.2 | 98.0 | 85.1 | 75.6 | 45.4 | 95.7 | 75.9 | |
| DriveSuprim (Yao et al., 2025) | ✓ | V2-99 | Stage 1 | 98.9 | 95.1 | 99.2 | 99.6 | 76.1 | 99.1 | 94.7 | 97.6 | 54.2 | 42.1 |
| | | | Stage 2 | 87.9 | 88.8 | 89.6 | 98.8 | 80.3 | 86.0 | 53.5 | 97.1 | 56.1 | |
| | | EVA-ViT-L | Stage 1 | 98.7 | 98.0 | 99.1 | 99.8 | 75.9 | 98.7 | 94.7 | 97.6 | 49.8 | 44.7 |
| | | | Stage 2 | 89.5 | 89.6 | 92.9 | 98.5 | 78.9 | 86.4 | 55.3 | 96.5 | 52.7 | |
| | | ViT-L | Stage 1 | 97.8 | 97.3 | 98.9 | 99.3 | 77.1 | 98.2 | 95.8 | 97.6 | 50.2 | 43.4 |
| | | | Stage 2 | 90.3 | 88.9 | 90.8 | 98.9 | 81.1 | 87.4 | 54.2 | 95.1 | 48.3 | |
| GTRS-Dense (Li et al., 2025f) | ✓ | V2-99 | Stage 1 | 98.7 | 95.8 | 99.4 | 99.3 | 72.8 | 98.7 | 95.1 | 96.9 | 40.4 | 41.7 |
| | | | Stage 2 | 91.4 | 89.2 | 94.4 | 98.8 | 69.5 | 90.1 | 54.6 | 94.1 | 49.7 | |
| | | EVA-ViT-L | Stage 1 | 97.6 | 95.8 | 99.7 | 99.8 | 77.2 | 97.8 | 95.3 | 97.3 | 46.7 | 43.4 |
| | | | Stage 2 | 91.9 | 91.3 | 92.7 | 99.0 | 72.7 | 90.4 | 53.8 | 94.1 | 41.6 | |
| | | ViT-L | Stage 1 | 98.9 | 98.2 | 99.8 | 99.6 | 73.9 | 98.9 | 95.3 | 97.3 | 40.0 | 45.3 |
| | | | Stage 2 | 91.5 | 90.8 | 94.7 | 98.5 | 70.8 | 90.1 | 55.4 | 97.2 | 54.2 | |
| ZTRS (Ours) | ✗ | ViT-L | Stage 1 | 98.6 | 96.7 | 99.8 | 99.8 | 72.1 | 98.0 | 95.6 | 97.6 | 51.6 | 45.0 |
| | | | Stage 2 | 88.9 | 90.9 | 94.6 | 97.9 | 70.8 | 87.1 | 58.6 | 97.5 | 63.6 | |
| | | V2-99 | Stage 1 | 98.9 | 97.6 | 100.0 | 100.0 | 66.7 | 98.9 | 96.2 | 96.7 | 44.0 | **45.5** |
| | | | Stage 2 | 91.1 | 90.4 | 95.8 | 99.0 | 63.6 | 89.8 | 60.4 | 97.6 | 66.1 | |

Table 2: **Performance on the Navtest Benchmark.**

| Method | IL | Backbone | NC↑ | DAC↑ | DDC↑ | TL↑ | EP↑ | TTC↑ | LK↑ | HC↑ | EC↑ | EPDMS↑ |
|---|---|---|---|---|---|---|---|---|---|---|---|---|
| Human Agent | - | - | 100 | 100 | 99.8 | 100 | 87.4 | 100 | 100 | 98.1 | 90.1 | 90.3 |
| Ego Status MLP | ✓ | - | 93.1 | 77.9 | 92.7 | 99.6 | 86.0 | 91.5 | 89.4 | 98.3 | 85.4 | 64.0 |
| Transfuser (Chitta et al., 2022) | ✓ | ResNet34 | 96.9 | 89.9 | 97.8 | 99.7 | 87.1 | 95.4 | 92.7 | 98.3 | 87.2 | 76.7 |
| HydraMDP++ (Li et al., 2025b) | ✓ | ResNet34 | 97.2 | 97.5 | 99.4 | 99.6 | 83.1 | 96.5 | 94.4 | 98.2 | 70.9 | 81.4 |
| | | V2-99 | 98.4 | 98.0 | 99.4 | 99.8 | 87.5 | 97.7 | 95.3 | 98.3 | 77.4 | 85.1 |
| | | ViT-L | 98.5 | 98.5 | 99.5 | 99.7 | 87.4 | 97.9 | 95.8 | 98.2 | 75.7 | 85.6 |
| DriveSuprim (Yao et al., 2025) | ✓ | ResNet34 | 97.5 | 96.5 | 99.4 | 99.6 | 88.4 | 96.6 | 95.5 | 98.3 | 77.0 | 83.1 |
| | | V2-99 | 97.8 | 97.9 | 99.5 | 99.9 | 90.6 | 97.1 | 96.6 | 98.3 | 77.9 | 86.0 |
| | | ViT-L | 98.4 | 98.6 | 99.6 | 99.8 | 90.5 | 97.8 | 97.0 | 98.3 | 78.6 | **87.1** |
| ZTRS (Ours) | ✗ | V2-99 | 97.8 | 99.4 | 99.8 | 99.8 | 84.1 | 97.0 | 96.2 | 98.2 | 77.2 | 85.3 |
| | | ViT-L | 98.2 | 99.1 | 99.7 | 99.8 | 86.9 | 97.5 | 96.6 | 98.2 | 78.2 | 86.2 |

Table 3: **Zero-shot Performance on the HUGSIM Benchmark.** *Official results from Zhou et al. (2024) on both public and unreleased private scenarios. The rest are based on the public scenarios.

| Method | Easy | | Medium | | Hard | | Extreme | | Overall | |
|---|---|---|---|---|---|---|---|---|---|---|
| | RC | HD-Score | RC | HD-Score | RC | HD-Score | RC | HD-Score | RC | HD-Score |
| UniAD (Hu et al., 2023)* | 58.6 | 48.7 | 41.2 | 29.5 | 40.4 | 27.3 | 26.0 | 14.3 | 40.6 | 28.9 |
| VAD (Jiang et al., 2023)* | 38.7 | 24.3 | 27.0 | 9.9 | 25.5 | 10.4 | 23.0 | 8.2 | 27.9 | 12.3 |
| LTF (Chitta et al., 2022)* | 68.4 | 52.8 | 40.7 | 24.6 | 36.9 | 19.8 | 25.5 | 8.1 | 41.4 | 24.8 |
| LTF (Chitta et al., 2022) | 60.4 | 42.5 | 39.4 | 17.7 | 32.7 | 11.8 | 27.9 | 10.6 | 37.9 | 18.0 |
| GTRS-Dense (Li et al., 2025f) | 64.2 | 55.5 | 50.0 | 39.0 | 20.7 | 11.7 | 22.3 | 14.3 | 38.0 | 28.6 |
| ZTRS (Ours) | 74.4 | 60.8 | 50.9 | 34.2 | 32.7 | 20.5 | 21.9 | 11.0 | **42.6** | **28.9** |

the real-world Navtest benchmark. ZTRS achieves better open-loop planning performance than Hydra-MDP++ (Li et al., 2025b) with both V2-99 and ViT-L backbones, but still falls behind DriveSuprim (Yao et al., 2025), which utilizes more advanced data augmentation techniques and scoring architectures. Finally, Tab. 3 shows the zero-shot performance on the HUGSIM benchmark. Without adaptation to simulated data or closed-loop driving, ZTRS achieves the best RC and HD-Score on public scenarios, outperforming IL-based GTRS-Dense by 4.6% RC and 0.3% HD-Score.

## 3.4 ABLATION STUDY

In Tab. 4, we study the effects of different learning paradigms and targets. When using the trajectory with the maximum EPDMS score (i.e., $\hat{\mathcal{E}} = \operatorname*{argmax}_{a} \mathcal{E}(s, a)$) as the imitation target, perfor-

Table 4: **Ablation study on different learning paradigms and targets.** $\hat{\mathcal{E}}$ represents using the trajectory with the maximum ground-truth EPDMS as the imitation target, while ll and log-ll represent optimization with likelihoods over all actions and the log-likelihood over a sampled action.

| IL | RL | Target | NC ↑ | DAC ↑ | DDC ↑ | TL ↑ | EP ↑ | TTC ↑ | LK ↑ | HC ↑ | EC ↑ | EPDMS ↑ |
|---|---|---|---|---|---|---|---|---|---|---|---|---|
| ✓ | ✗ | Human | 98.5 | 98.7 | 98.9 | 99.9 | 88.5 | 98.2 | 97.0 | 98.3 | 80.5 | 86.2 |
| ✓ | ✗ | $\hat{\mathcal{E}}$ | 96.6 | 96.8 | 99.5 | 99.6 | 88.3 | 96.0 | 96.7 | 92.2 | 18.5 | 76.7 |
| ✗ | ll | $\mathcal{E}$ | 97.5 | 99.2 | 99.8 | 99.7 | 89.3 | 96.9 | 96.8 | 98.0 | 53.8 | 84.2 |
| ✗ | ll | $\mathcal{E} - b$ | 97.8 | 99.4 | 99.8 | 99.8 | 84.1 | 97.0 | 96.2 | 98.2 | 77.2 | **85.3** |
| ✗ | log-ll | $\mathcal{E} - b$ | 97.7 | 96.7 | 99.7 | 99.9 | 73.0 | 96.1 | 93.2 | 97.7 | 36.1 | 75.0 |

Table 5: **The relationship between the size of the action space and evaluation data.** $\text{EPDMS}_1$ measures the real-world portion of Navhard, while $\text{EPDMS}_2$ measures the simulated portion.

| Backbone | $|\mathcal{A}|$ for training | $|\mathcal{A}|$ for inference | Navtest | Navhard | | |
|---|---|---|---|---|---|---|
| | | | EPDMS | $\text{EPDMS}_1$ | $\text{EPDMS}_2$ | EPDMS |
| V2-99 | 8192 | 8192 | 84.6 | 73.3 | 57.4 | 43.0 |
| | 16384 | 16384 | **85.3** | **74.9** | 57.1 | 43.4 |
| | 16384 | 8192 | 82.0 | 74.2 | **60.7** | **45.5** |
| ViT-L | 8192 | 8192 | 84.6 | 73.7 | 55.9 | 41.9 |
| | 16384 | 16384 | **86.2** | **76.1** | 50.5 | 38.8 |
| | 16384 | 8192 | 84.3 | 73.4 | **59.9** | **45.0** |

mance drops significantly compared to the IL baseline, as many trajectories in $\mathcal{A}$ can achieve high EPDMS and a single target fails to capture the underlying pattern. Using likelihoods over the entire action space mitigates this issue, improving EPDMS by 7.5%, but introduces serious oscillation, as indicated by the low EC metric. This highlights the need for our correction term $b$ to enforce temporal consistency. Using $\mathcal{E} - b$ as the reward increases EC by 23.4%. In contrast, computing log-likelihoods over a sampled action fails to perform equally, which demonstrates the effectiveness of our EPO method in providing dense supervision for the entire action space.

Tab. 5 shows the relationship between the size of the action space and evaluation data. We observe that models using the full action space during inference achieve the best results on real-world data, as reflected by EPDMS on Navtest and Navhard, while shrinking the action space improves performance on the simulated portion. This finding is consistent with GTRS (Li et al., 2025f): regardless of whether the model is trained with human demonstrations, reducing model complexity tends to enhance generalization on unseen simulated data.

### 3.5 QUALITATIVE RESULTS.

Fig. 3 and Fig. 4 show visualization results on the open-loop planning benchmark Navtest and the closed-loop driving benchmark HUGSIM, respectively. Interestingly, ZTRS learns driving patterns that resemble human trajectories from rule-based rewards, though it is trained without human demonstrations. Moreover, ZTRS manages to navigate safely in safety-critical driving scenarios without training on simulated data, as shown in Fig. 4. Even under the extreme condition depicted in Fig. 4 (c), where the ego agent must overtake a parked car while facing an oncoming vehicle, ZTRS can safely complete the route. This demonstrates the strong closed-loop driving ability of ZTRS.

## 4 RELATED WORK

### 4.1 END-TO-END IMITATION LEARNING FOR AUTONOMOUS DRIVING

Given an offline expert dataset, Imitation Learning (IL) trains a policy to mimic expert behavior. In the end-to-end autonomous driving literature (Chen et al., 2024a), early IL-based approaches (Codevilla et al., 2018) proved effective in the closed-loop driving simulator

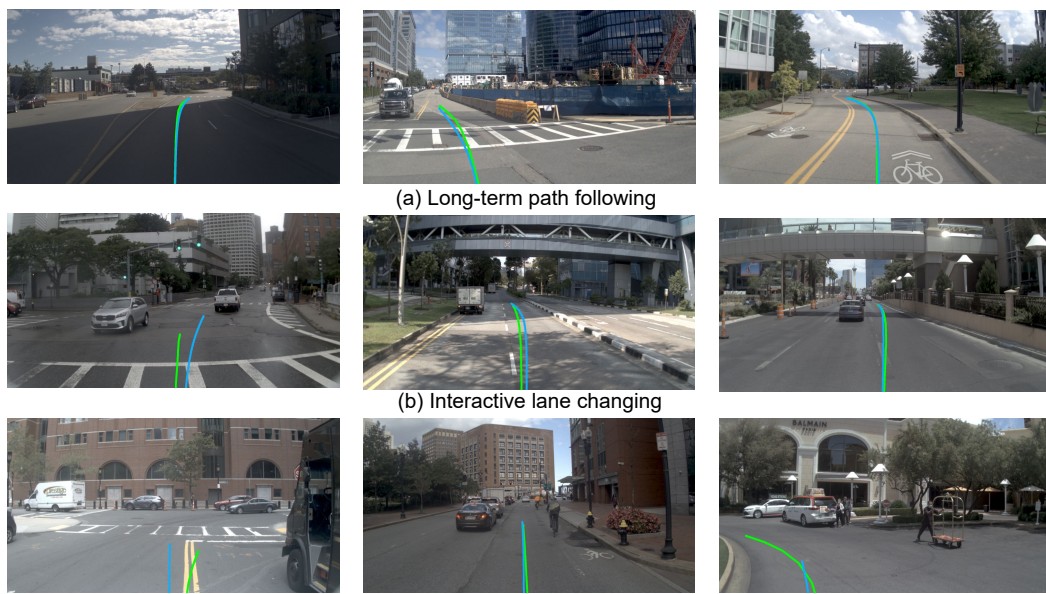

(a) Long-term path following

(b) Interactive lane changing

(c) Cautious driving near parked vehicles and pedestrians

Figure 3: **Visualizations of planned trajectories (blue curves) and the human trajectory (green curves) on the open-loop planning benchmark Navtest.**

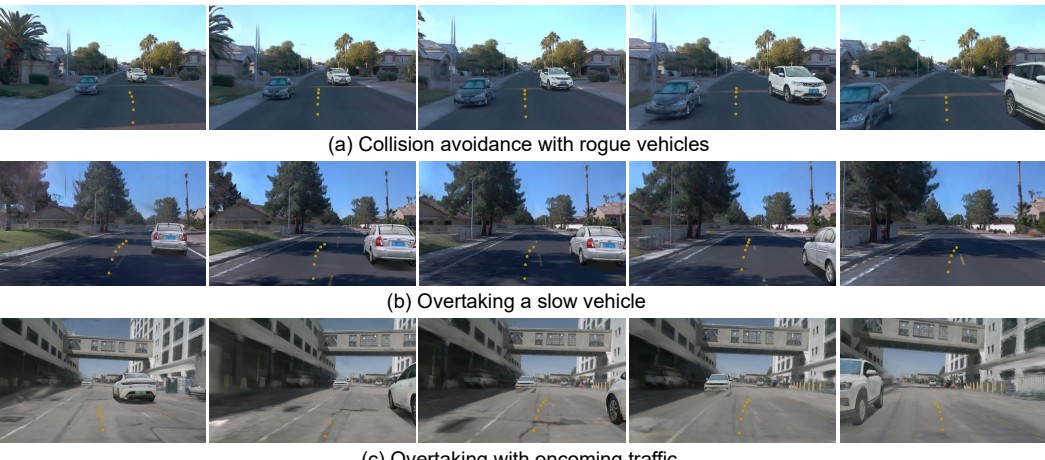

(a) Collision avoidance with rogue vehicles

(b) Overtaking a slow vehicle

(c) Overtaking with oncoming traffic

Figure 4: **Visualizations of planned trajectories (orange dots) on the challenging closed-loop driving benchmark HUGSIM.**

CARLA (Dosovitskiy et al., 2017). Subsequent works improve the closed-loop driving performance with modern neural architectures (e.g. Transformers (Vaswani et al., 2017)) (Chitta et al., 2022), intermediate representations (Hu et al., 2022; Renz et al., 2022; Shao et al., 2023; Jia et al., 2023b), and policy distillation (Chen et al., 2020; Zhang et al., 2021; Wu et al., 2022; Jia et al., 2023a). The introduction of UniAD (Hu et al., 2023) highlighted the strength of IL on real-world sensor data, whose complexity and diversity greatly exceed synthetic data produced by simulators. Building on this foundation, numerous efforts further focus on efficiency (Jiang et al., 2023; Liao et al., 2025), multi-modal behaviors (Chen et al., 2024b; Liao et al., 2025), vision-language understanding (Wang et al., 2024; Li et al., 2025d), and safety constraints (Li et al., 2024b).

## 4.2 Reinforcement Learning for Autonomous Driving

Unlike Imitation Learning, Reinforcement Learning (RL) (Sutton et al., 1998) trains agents to maximize rewards by interacting with the environment, and autonomous driving RL methods generally fall into two categories: symbolic-input methods and sensor-based methods, both relying on simulators. Symbolic-input methods (Toromanoff et al., 2020; Zhang et al., 2021; Li et al., 2024a; Cusumano-Towner et al., 2025; Jaeger et al., 2025) use low-dimensional abstractions (e.g. 3D bounding boxes, maps, traffic signals) and have shown strong results on CARLA (Dosovitskiy et al., 2017), nuPlan (H. Caesar, 2021), and Waymax (Gulino et al., 2023). GigaFlow (Cusumano-Towner et al., 2025) and CaRL (Jaeger et al., 2025) both demonstrated that large-scale RL could train robust policies from scratch. On the other hand, sensor-based approaches operate on raw inputs like images. Early attempts (Kendall et al., 2019) explored real-world training, but faced safety and efficiency challenges, while subsequent works (Nehme & Deo, 2023; Delavari et al., 2025; Yang et al., 2025) relied on simulated sensor data in CARLA. Despite success in simulators, such approaches could face sim-to-real gaps. Recently, RAD (Gao et al., 2025) fine-tuned a policy in a 3DGS simulator with high-dimensional real-world images, but still depended on human demonstrations for pre-training and reward computation. Similarly, several other approaches avoid the cold-start problem by fine-tuning an IL-pretrained diffusion policy (Li et al., 2025d;a). In contrast, our approach learns trajectory planning with reward signals from scratch, while operating on high-dimensional real-world sensor data.

## 4.3 Scoring-based End-to-end Trajectory Planning

The scoring-based end-to-end planning method introduces a predefined vocabulary containing multiple trajectories, and scores each trajectory to select the most appropriate candidate as the output. Early works (Philion & Fidler, 2020; Phan-Minh et al., 2020; Chen et al., 2024b) score the trajectory candidates through classification based on their distance towards the ground-truth human trajectory. Beyond relying on a single human demonstration, the Hydra-MDP series (Li et al., 2024b; 2025b) proposed multi-target hydra-distillation to score trajectories with multiple rule-based metrics, leading to more robust planning capability. SafeFusion (Wang et al., 2025) synthesizes collision-related scenarios for training a robust planning model and eases the reliance on imitation learning. Other works further introduce multiple approaches to reach more precise and comprehensive trajectory scoring, such as test-time training (Sima et al., 2025), iterative refinement (Yao et al., 2025), and diffusion-based trajectory generation (Li et al., 2025e;f). Despite these advancements, these scorers still rely heavily on the imitation of human trajectories. In contrast, our approach fully discards expert demonstrations and achieves strong end-to-end planning performance via reinforcement learning.

## 5 Conclusion

We present ZTRS, the first end-to-end autonomous driving framework that eliminates imitation learning. In summary, ZTRS demonstrates that end-to-end planners can be trained entirely without human demonstrations by leveraging offline data, reward-driven supervision, and Exhaustive Policy Optimization. By densely optimizing over enumerable actions, ZTRS overcomes the cold-start problem and achieves robust planning by operating on high-dimensional sensor inputs. Extensive evaluations on real-world and simulated benchmarks show that ZTRS not only matches or exceeds IL-based planners in planning capabilities but also establishes new state-of-the-art performance in challenging and safety-critical conditions. These results highlight the potential of relying on rewards rather than human demonstrations to achieve reliable end-to-end autonomous driving.

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
