# OpenReview forum: "ZTRS: Zero-Imitation End-to-end Autonomous Driving with Trajectory Scoring"
_ICLR.cc/2026/Conference — ICLR 2026 Conference Withdrawn Submission_

### Official Review · Reviewer_qN45 · 2025-10-29

**Soundness:** 3
**Presentation:** 3
**Contribution:** 2
**Rating:** 4
**Confidence:** 5

**Summary:**

This paper introduces ZTRS, a novel framework for autonomous driving that bypasses traditional imitation learning by directly training from sensor data using reinforcement learning.  ZTRS leverages a new technique called Exhaustive Policy Optimization to handle complex actions and rewards, enabling robust planning even with raw sensor inputs. The system demonstrates state-of-the-art performance on challenging benchmarks, showcasing its potential for end-to-end autonomous driving.

**Strengths:**

* End-to-end solution for Autonomous Driving
  * Uses differentiable autonomous driving stack without VLM
  * Uses Reinforcement Learning w/o IL where the advantage is defined by EPDMS $\varepsilon$ with an optional correction term $b$
    * Done by so called EPO which is a version of the policy gradient for aoffline data and enumerable actions

**Weaknesses:**

* No mentioning of CIMRL work [1] that also tries to learn RL-based trajectory scorer on top of any trajectory source (could be either IL or RL-based); it seems like a good reference fit in the main idea of the paper + Section "4.2 RL For AD", symbolic-input methods
* No rigorous ablations on rewards (what to include and what to exclude), and sampling over the sum of $m+1$ trajectories (Section 2.1)
* Unclear why weight decay factor is equal to 0.0, not to a small number
* The results using the closed-loop evaluation (Table 3) doesn't look very exciting

[1] Booher, Jonathan, et al. "Cimrl: Combining imitation and reinforcement learning for safe autonomous driving." arXiv preprint arXiv:2406.08878 (2024).

**Questions:**

* Table 1 and Table 2 seem a little bit inconsistent - why the set of baselines is not the same as it is an open-loop comparison?

---

### Official Review · Reviewer_NvzM · 2025-10-30

**Soundness:** 2
**Presentation:** 2
**Contribution:** 2
**Rating:** 2
**Confidence:** 4

**Summary:**

This paper introduces ZTRS (Zero-Imitation End-to-End Autonomous Driving with Trajectory Scoring), a framework for end-to-end autonomous driving that operates on raw sensor inputs. The central claim is that ZTRS is "zero-imitation," meaning it is trained entirely with offline reinforcement learning (RL) using rule-based rewards (EPDMS), without any reliance on human expert demonstrations for Imitation Learning (IL). The model adopts a "trajectory scorer" architecture, where a Transformer-based model learns to select the best trajectory from a pre-defined, discrete vocabulary of candidate trajectories. The authors propose a policy gradient variant called "Exhaustive Policy Optimization" (EPO) to train this policy by densely optimizing over the entire enumerable action space. The framework is evaluated on Navtest, Navhard, and HUGSIM, claiming state-of-the-art results on the Navhard benchmark.

**Strengths:**

1. The goal of reducing or eliminating the dependency on large-scale, high-quality expert demonstrations (IL) is a significant and well-motivated problem in autonomous driving.

2. The paper presents strong results on the challenging Navhard benchmark, outperforming prior IL-based methods. This demonstrates that the proposed RL-based approach can be effectively optimized for the given offline metrics.

**Weaknesses:**

1. The "trajectory scorer" paradigm is not new and has been extensively used by many prior works cited in this paper (e.g., Hydra-MDP, GTRS-Dense, DriveSuprim). The overall architecture (image backbone, trajectory tokenizer, transformer decoder) is standard for this class of models.

2. The method presented as EPO is a direct and standard application of the Policy Gradient Theorem for a discrete, enumerable action space.  When the action space is small enough to enumerate, computing the full sum is the most direct implementation, not a novel optimization technique.

3. The entire learning signal is derived from the EPDMS score, a complex, hand-engineered proxy for "good driving." The framework is essentially learning to "hack" this specific metric. It is not surprising that it performs well on a benchmark (Navhard) that uses this same metric for evaluation. This does not guarantee generalizable or robust driving intelligence, but rather a policy that is highly overfitted to a specific, non-learnable reward function.

**Questions:**

1. Given the framework's reliance on a fixed, non-generative trajectory vocabulary, how is it expected to handle novel or "long-tail" scenarios where the safe and optimal trajectory is not present (or even approximated well) within the 16k candidate set?

2. The strong performance on Navhard seems tightly coupled with the fact that the reward signal (EPDMS) is also the evaluation metric. How can the authors be sure the model is learning a generalizable driving skill rather than simply overfitting to the specific rules of the EPDMS score?

---

### Official Review · Reviewer_D3rq · 2025-11-01

**Soundness:** 2
**Presentation:** 2
**Contribution:** 2
**Rating:** 4
**Confidence:** 3

**Summary:**

This paper presents an end-to-end autonomous driving framework. The goal is to alleviate issues from Imitation Learning like the need for high-quality expert demonstrations and covariate shift. The paper proposes Reinforcement Learning on real data inputs and giving rewards based on an automatic scorer of actions, instead of learning a policy in a simulator. Results are shown on several dataset like Navtest, Navhard and Hugsim.

**Strengths:**

The figures are very clear and explain the arguments and descriptions well.

In general the idea to score the output to get rewards for a Reinforcement Learning approach is good for the autonomous driving domain.

The approach was tested on a larger range of benchmarks and seems to be close or at the state-of-the-art.

Useful ablation studies, e.g. on the size of the action space, allow to estimate the robustness of the approach.

**Weaknesses:**

The Exhaustive Policy Optimization seems to have strong similarities with Group Relative Policy Optimization but calling it differently. The authors should either consider framing their approach within the context of GRPO or explain how it is different in the related work.

The language of the paper could be improved. The conclusion states that the approach "eliminates" imitation learning. This is not something which can be expected from any single approach in the near future nor a statement with a concrete enough meaning.

The tables make it very hard to see what the best results for the individual metrics are. It seems like the approach is near or at state-of-the-art but not by a large margin.

Having related work before the conclusion is a possible location but seems uncustomary in this domain. I'd suggest to put it after the introduction.

In summary, it seems the contribution does not improve upon state-of-the-art by a large margin and potentially re-invent GRPO. However, this is not entirely clear. The general idea of using metrics on the actions for reinforcement learning of a driving policy is good so I see this as a borderline paper which could polish and make its main contribution clearer and should ideally improve its results but it may as well be accepted.

**Questions:**

Please highlight the best results for individual metrics like NC as bold in Table 1, 2 and everywhere else where this makes sense. In table 5 this was well done. Furthermore, Table 2 has useful arrows showing if a metric should be minimized or maximized. Please add this to table 1.

Please explain the relationship between exhaustive policy optimization and GRPO.

---

### Note · Authors · 2025-11-12

I have read and agree with the venue's withdrawal policy on behalf of myself and my co-authors.